# Halogen-Free Flame Retardant Polypropylene Fibers with Modified Intumescent Flame Retardant: Preparation, Characterization, Properties and Mode of Action

**DOI:** 10.3390/polym13152553

**Published:** 2021-07-31

**Authors:** Qibin Xu, Lei Wu, Xiang Yan, Shengchang Zhang, Linan Dong, Zexi Su, Tianhaoyue Zhong, Chunhui Jiang, Yuan Chen, Mengjin Jiang, Pengqing Liu

**Affiliations:** 1College of Polymer Science & Engineering, Sichuan University, Chengdu 610065, China; xuqibin2015@163.com (Q.X.); shuhliang@163.com (L.W.); yanxiang2020@foxmail.com (X.Y.); zhangshengc24@126.com (S.Z.); zexisu@163.com (Z.S.); zthy5158@163.com (T.Z.); jiangchui2021@163.com (C.J.); cyuan201431@163.com (Y.C.); memoggy@126.com (M.J.); 2Chongqing Academy of Metrology and Quality Inspection, Chongqing 401120, China; bobleec1@163.com

**Keywords:** polypropylene fibers, melt spinning, phosphorus-nitrogen compound, halogen-free, intumescent flame retardant, mode of action

## Abstract

A novel intumescent flame retardant (IFR) agent designated as Dohor-6000A has been used to prepare halogen-free flame retardant polypropylene (PP) fibers via melting spinning. Before being blended with PP resin, a surface modification of Dohor-6000A was carried out to improve its compatibility with the PP matrix. The rheological behavior of flame retardant Dohor-6000A/PP resin, the structure, morphology, mechanical properties, flammability of the Dohor-6000A/PP fibers were studied in detail, as well as the action mode of flame retardant. X-ray diffraction (XRD) showed that the addition of Dohor-6000A did not damage the crystal as well as the orientation structure of PP matrix, which was helpful to the maintenance of mechanical properties. The presence of the IFR significantly improved the flame retardant performance and thermal stability of PP fibers. When the content of Dohor-6000A reached 25%, the fibers displayed a limiting oxygen index (LOI) value of 29.1% and good melt-drop resistance. Moreover, the peak heat release rate (PHRR) and total heat release (THR) from microscale combustion colorimetry (MCC) tests were decreased by 26.0% and 16.0% in comparison with the same conditions for pure PP fibers. In the condensed phase, the IFR promoted a carbonization process and promoted the formation of a glassy or stable foam protective layer on the surface of the polymer matrix. In addition, the IFR decomposed endothermically to release of non-combustible gases such as NH_3_ and CO_2_ which dilutes the combustible gases in the combustion zone.

## 1. Introduction

Polypropylene (PP) fibers are produced from the melt spinning of isotactic PP [1]. As one of the main fiber varieties, due to their low density, low moisture absorption rate, high toughness, good thermal insulation, antibacterial, chemical stability, and good antifouling, PP fibers are widely used in interior decorations and industrial fields [2,3,4,5,6]. Although PP fiber has many excellent properties, high flammability still limits its further applications and development [7,8]. Its limiting oxygen index (LOI) is only about 18% [9]. The decomposition products during PP combustion are also combustible, and a large number of molten droplets are generated, which causes the fire to spread rapidly [10,11,12,13,14]. Therefore, the demand for the flame-retarding modification of PP is exigent [15].

Investigations on the flame-retardant modification of PP fibers have experienced three main stages: the use of halogen-containing [14], low-halogen, and halogen-free agents, respectively. Organohalogen flame retardants usually release volatile hydrogen halides formed from the interactions with thermally degraded polymers into the gas phase, and then trap the radicals that accelerates the combustion process [16,17,18]. However, organohalogen flame retardants produce potentially carcinogenic toxic substances such as furans and dioxins during their combustion processes [18,19]. More importantly, the discarded items containing organohalogen flame retardants not only bring pollution to environment, but also endanger human’s health via bio-accumulation and then transferring into food chain [18,20,21]. Human exposure may lead to the generation of several diseases [17,18]. Therefore, efforts have been made to reduce and even avoid the use of halogen-containing compounds to achieve flame-retardant effects [22,23]. For industrial production, combinations of flame retardants are often used [24]. Polypropylene fibers have been modified with halogen-free inorganic flame retardants such as carbon nanotubes [25], metal hydroxide flame retardants, or phosphate or polyphosphate [26] to help improve flame retardancy. A mixture of clay, ammonium polyphosphate, and a hindered amine has been used to improve the flame retardancy of polypropylene fabric [27]. Such flame retardants must be present in large amounts to achieve appropriate flame retardant effects. For inorganic flame retardants, more than 60% in polypropylene is required to achieve satisfactory flame retardancy [28]. For single or mixed metal hydroxide flame retardants, about 60 wt% and 30 wt%, respectively, is needed to achieve a good flame retardant effect. The presence of a high amount of flame retardant significantly reduces the mechanical properties of polypropylene especially in the case of surface migration or uneven dispersion [29,30]. Because of its low thermal stability in the processing temperature range for polypropylene, polyphosphate finds limited application in textiles [26].

Compared with inorganic flame retardants, intumescent flame retardants (IFR) have a higher flame-retardant efficiency such that a lower amount is required to achieve a certain flame-retardant effect [31,32]. IFR is an environmentally friendly flame retardant that may be generated from phosphorus-nitrogen compounds. A typical IFR includes three parts: a carbon source, an acid source, and a nitrogen source [33,34]. IFR may have good flame-retardant effects in both the condensed-phase and the gas-phase [35,36,37,38]. IFR is generally active in the solid phase to produce an expanded char layer at the surface of the substrate. This acts as an insulation barrier to inhibit heat feedback from the combustion zone which decreases the rate of polymer degradation to form volatile fuel fragments to feed the combustion process [18,39]. This may also suppress the generation of smoke and promote melt-drop resisstance [30,40]. When thermaloxidation processes are involved in polymer decomposition, the presence of oxygen at the surface is important. Inert gases formed from decomposition of the blowing agent may dilute the fuel load in the gas phase. However, fiber polymer treated with IFR has poor spinnability and it is difficult to obtain fibers with good mechanical properties due to the poor compatibility of the additives with the polypropylene matrix [41,42,43]. It is possible to modify an IFR to improve combability with the polymer matrices such as acrylonitrile-butadiene-styrene [44] and polyester [45]. However, there are few reports on the application of modified IFR in polypropylene fibers for fabric.

In the work reported here, a special IFR for polypropylene, Dohor-6000A, was mechanically modified to make it finer and enhance its compatibility with polypropylene, to meet the requirements of melt spinning. Furthermore, the surface of Dohor-6000A was modified using a compatibilizer to enhance the dispersion and compatibility with the polypropylene matrix. Halogen-free flame retardant polypropylene fibers were prepared by melt spinning. The aggregation structure, morphology and mechanical properties of the fibers were characterized using differential scanning calorimeter (DSC), X-ray diffractometer (XRD), two-dimensional wide-angle diffractometer (WAXD), scanning electron microscope (SEM), polarizing microscope (POM), and electronic single yarn strength meter. Therefore, the influence of the presence of the flame retardant on the crystallization behavior, fiber morphology and mechanical properties of polypropylene fibers was clarified. SEM, LOI, thermal gravimetric analysis (TG), microscale combustion colorimetry (MCC), and thermal gravimetric analysis-Fourier transform infrared spectroscopy (TG-FTIR) were used to investigate the effect of the IFR on the flame retardant properties of polypropylene fibers and systematically probe the flame retardant mode of action. The costs and availability of raw materials combined with commonly used suggests that this manufacturing method could represent a new approach for preparing halogen-free flame-retardant polypropylene fiber.

## 2. Experimental

### 2.1. Materials

Polypropylene resin (T30S, MFI = 2.6 g/10 min) was sourced from Lanzhou Petrochemical Co., Ltd., Lanzhoou, China. Maleic anhydride grafted polypropylene (E43) was purchased from Westlake Chemical Co., Ltd., Houston, TX, USA as the compatibilizer. ϒ-Aminopropyltriethoxysilane (KH-550), glacial acetic acid and absolute ethanol were obtained from Chengdu Kelong Chemical Reagent Factory, Chengdu, China. Dohor-6000A (powder, average size < 7 μm) was obtained from Dohor New Material Technology Co., Ltd., Dongguan, China. Dohor-6000A is a blended nitrogen-phosphorus IFR and the mass contents of the phosphorus, nitrogen and carbon are 19.78%, 21.31% and 17.45%, respectively. Deionized water was self-made in laboratory.

### 2.2. Modification of the Dohor-6000A

Coupling modifier containing 20 wt% KH-550, 78 wt% ethanol and 2 wt% distilled water was used to modify the surface of Dohor-6000A. After mixing these three reagents, the pH was adjusted to 4-5 with glacial acetic acid. Then, the prepared modifier was put into the ultrasonic machine for several minutes to make the solution evenly dispersed. Next, the modifier was added to Dohor-6000A and mixed via a high-speed pulverizer for several minutes to perform mechanical surface modification. Afterwards, the modified Dohor-6000A with 2 wt% coupling modifier was obtained after drying for 2 h.

### 2.3. Preparation of Halogen-Free Flame Retardant Polypropylene Chips

Melt extrusion was used to prepare polypropylene chips for melt spinning. The polypropylene resin, the MA-g-polypropylene and the modified flame retardant agent with different ratios were blended in a twin-screw extruder and granulated into chips. The blending compositions were shown in Table 1. The mass content of flame retardant is from 0% to 30%, and that of compatibilizer maleic anhydride grafted polypropylene is 0% or 5%.

### 2.4. Melt Spinning Halogen-Free Flame Retardant Polypropylene Chips

Polypropylene fibers were produced via melt spinning by a melt-spinning machine (Beijing Paigu Precision Machinery Co., Ltd., Beijing, China). First, polypropylene pellets were introduced into the feeding zone of a single-screw extruder and heated with five heating zones. The volumetric pump forced the molten polymer pellets toward the spinneret with 18 orifices. The as-spun fibers were obtained at the take-up speed of 200 m min^−1^. Then, the post-drawing and heat-setting process was carried out for the as-spun fibers to endow the fibers with higher orientation. The post-treatment conditions were as follows: stretching ratio is 4, stretching temperature is 130 °C, heat-setting time is 300 s.

### 2.5. Characterization

The morphology of the flame retardant, the fibers and the residual carbon were examined using an Inspect F50 field emission scanning electron microscope (Field Electron and Ion Co., Hillsboro, OR, USA) at an acceleration voltage of 20 kV. The SEM instrument was integrated with an energy dispersive X-ray (EDX) microanalyzer for elemental analysis.

The particle size and distribution of the flame retardant were measured with a Masterizer 2000 laser particle size analyzer (Malvern Instruments Ltd., Malvern, UK). The test range was 0.02–200 μm and the scanning speed was 1000 times s^−1^.

The rheological behavior of the polypropylene blends at 230 °C was evaluated via an Adopt RH7D high-pressure capillary rheometer (Malvern Instruments Ltd., Malvern, UK). The capillary had 1 mm diameter and 16 mm long (L/D = 16). Computer-aided monitoring records the shear rate and the pressure difference across the capillary, during the process from preheating to extrusion. In detail, the corresponding shear stress τ and apparent viscosity ƞ_a_ were obtained through computer data processing.

The thermal behavior of the fibers was recorded using a 204F1 type differential scanning calorimeter (Netzsch Instrument Manufacturing Co., Ltd., Selb, Germany). The scanning process includes initial heating at a rate of 10 °C min^−1^ from 30 to 200 °C and then cooled at a rate of 10 °C min^−1^. The crystallization exotherm and the melting endotherm were recorded. The crystallinity index (X_c_) was calculated according to Equation (1):(1)$$X_{c}\%=\frac{\Delta H_{m}}{\left(1-w_{m}\right)\times \Delta H_{m}^{0}}\times 100\%$$
where Δ*H*_m_ is the specific melting heat, $\Delta H_{m}^{0}$ is the theoretical specific melting heat of 100% crystalline isotactic polypropylene as 207 J g^−1^, and w_m_ is the weight fraction of IFR [46].

The crystal morphology of the fibers was observed using an Olympus BX-51 polarizing microscope equipped with a DP 27 CCD. The fiber samples were melted at 200 °C on the Linkam GS 350 hot stage, then cooled to 120 °C at 10 °C min^−1^.

XRD patterns were obtained with X’ Pert Pro wide-angle X-ray diffractometer (Philips Electronics Ltd., Netherlands) by Cu Kα radiation (λ = 1.5406 Å) at a scanning rate of 0.01° per second in the 2θ range of 4–45°. The Origin data analysis software was used to integrate the graphs obtained from the experiment. Then, its peak value was analyzed to find out the peak width at half maximum (H) and relative peak intensity. The orientation factor of the crystal region (f) was calculated according to Equation (2):(2)$$f=\frac{180-H}{180}$$

The tensile properties were measured using an YG061 electronic single yarn strength meter (Laizhou Electronic Instrument Co., Ltd., Laizhou, China). The test length was 20 mm, and the deformation rate was 10 mm min^−1^.

The LOI values were measured using a JF-3 oxygen index measuring instrument (Jiangning District Analytical Instrument Co., Ltd., China) according to ISO-4589-2. The fibers were braided into 6 mm × 3 mm × 70 mm bundles. The fibers were grinded into powders for the flammability tests. The test is conducted by a FTT0001 microscale combustion colorimeter (FTT Co., Ltd., East Grinstead, UK). About 5 mg of the samples were heated to 900 °C at a heating rate of 1 °C s^−1^ with nitrogen flow at 80 cm^3^ min^−1^.

The thermogravimetric analysis was performed by a Q500 thermal analyzer (TA Instrument Co., Newcastle, DE, USA) at a heating rate of 10 °C min^−1^ from 30 to 700 °C in an air atmosphere.

The gas molecules produced by the fibers during thermal decomposition were characterized by TG-FTIR. The sample was decomposed with the aid of TGA Q600 thermal weight loss instrument (TA Instrument Co., Newcastle, DE, USA) at a heating rate of 10 °C min^−1^ from 150 to 800 °C in a nitrogen atmosphere. The decomposition products were introduced into the Nicolet Magna IR 560 Fourier Transform Infrared Spectrometer (Nicolet Corporation, Madison, WI, USA). The resolution was 2 cm^−1^ and the scanning wave number range was 400–4000 cm^−1^.

## 3. Results and Discussion

### 3.1. Rheological Behavior of the Flame Retardant Polypropylene Resins

The high-pressure capillary rheological curves of the polypropylene blends at 230 °C are illustrated in Figure 1. The melt with lower flame retardant content has lower viscosity than pure polypropylene melt at a low shear rate. It is ascribed that the flame retardant is well dispersed in the polypropylene matrix, which hinders the entanglement of matrix resin molecules [47]. The density of entanglement points between molecular chains decreases, resulting in a decrease in the initial apparent viscosity, which acts as a lubricant. When the flame retardant content is up to 20%, the apparent viscosity of the melt is greater than that of pure polypropylene melt. The reason is that with the high content of the flame retardant, in addition to the lubricating effect, it also exists strong interaction with the polymer molecular chain which reduces the free volume of the polymer melt. It increases the steric resistance of the molecular chain and the apparent viscosity of the polymer melt. This phenomenon is also found and explained in some literatures concerning filled composite materials [47].

When the shear rate reaches a high value (e.g., 1000 s^−1^), the viscosity of all melts tends to be consistent, regardless of the content of the flame retardant, which represents similar fluidity of the melts. The reason is that the entanglement of molecular chains is almost completely untied under high shear rate. The shear rate of applied on the melt from the spinning orifice is much higher than 1000 s^−1^. Therefore, the apparent viscosity of the melt is in a stable state, no matter if IFR is introduced. This can ensure the stability of spinning of each sample.

### 3.2. Structure and Properties of the Flame Retardant Polypropylene Fibers

#### 3.2.1. Aggregation Structure

The DSC curves of the fiber samples are shown in Figure 2 and some typical data are collected in Table 2. It can be seen that the crystallization temperature (T_c_), crystallization enthalpy (ΔH_c_), and crystallinity (X_c_) of flame retardant polypropylene fibers are higher than these of pure polypropylene fiber. The fine flame retardant acts as a nucleating agent during the crystallization process of the matrix. These nucleating agents can reduce the occurrence of excessive cooling, thereby inducing the crystallization of the polypropylene matrix at higher temperatures. Flame retardant makes polypropylene heterogeneous nucleation, thereby increasing the nucleus density and crystallization rate of polypropylene. It can be found from the data of the full width at half maximum (W_1/2_) of the crystallization peak that the W_1/2_ of the flame retardant fibers is less than that of pure polypropylene fiber, which confirms that the modified Dohor-6000A increases the crystallization rate and reduces the crystal particle size. The crystallinity of the fiber with 30% flame retardant is less than that of the fiber with 10% flame retardant. This is because the excessively high content of flame retardant causes agglomeration phenomenon, which makes the particle size of some flame retardants larger. As a result, no more crystal nuclei can be formed and the growth of crystals is hindered, resulting in a decrease in the number of crystals and a decrease in crystallinity.

Figure 3 shows the polarizing microscope photographs of various polypropylene fiber samples. The size of spherulites formed by pure polypropylene fiber is significantly larger than that of other flame-retardant polypropylene fibers. The boundaries between the spherulites formed by pure polypropylene fiber are clear. As the amount of the IFR increases, the crystal size of flame retardant polypropylene fibers gradually decreases and the spherulite boundary becomes blurred. This behavior further confirms that the flame retardant plays a role of heterogeneous nucleation during the crystallization of polypropylene, which makes the spherulites smaller. In addition, the addition of flame retardant restricts the movement of polypropylene molecular chains. When the spherulite grows to a certain extent, the polypropylene molecular chain cannot quickly diffuse to the front of the spherulite growth due to the binding effect. This leads to imperfect spherulite growth and blurred spherulite boundaries.

XRD spectra of different fibers are shown in Figure 4. For pure polypropylene fiber, there are strong diffraction peaks at 2θ of 14.2°, 17.1°, and 18.6°, corresponding to the crystal planes (110), (040), and (130), respectively. These crystal planes belong to the unique crystal structure of α-polypropylene, indicating that only the α crystal form exists in pure polypropylene fiber. When the flame retardant is incorporated, the diffraction angle and the number of peaks in the XRD spectrum of the flame retardant fibers does not change. This shows that the addition of the IFR will not change the crystalline form of polypropylene fibers. However, the peak value of the diffraction peak decreased. It means the addition of the IFR leads to imperfect spherulite growth.

Figure 5 is 2D-WAXD diagrams of different fibers from polypropylene with different ratios of IFR. It can be seen that the polypropylene fibers with different content of IFR all have three kinds of symmetric diffraction bright spots, which represents the orientation of the crystal regions. It shows that the crystal area of each fiber sample has a certain orientation, and the addition of flame retardant does not change the orientation of the crystal area. The 2D-WAXD diffraction peak data of each fiber sample is shown in Table 3. As the amount of the IFR increases, the peak width at half maximum of the diffraction peak and the average orientation degree of the crystal region does not change significantly. On the basis of the analysis above, the addition of the IFR will not significantly damage the orientation of crystalline area for the fibers, which is good for mechanical properties of the fibers. 

#### 3.2.2. Morphology

The surface and cross-sectional morphologies of the fiber samples are shown in Figure 6. The surface of the pure polypropylene fiber is smooth, the cross section is dense, and there are no obvious defects. With the addition of the flame retardant, the surface of the fibers is relatively rough. This is because some flame retardant migrates to the surface of the fibers. When the content of flame retardant reaches 10 wt%, some grooves appear on the surface of the fiber and the cross section of the fiber is loose, with small defects. As the flame retardant content increases, this phenomenon is further enhanced. On the one hand, when the flame retardant content is too high such as 20 wt%, the flame retardant particles will agglomerate into particles with larger size, which causes defects inside the fibers. On the other hand, the performance of the IFR particles to shrink and deform during stretching is different, resulting in defects in the fibers. These grooves and defects play as stress concentration points, negatively affecting the mechanical properties of the fibers. From the fiber cross section, it can be seen that the IFR is evenly dispersed in polypropylene and there are no clear boundaries between the flame retardant particles and the matrix.

#### 3.2.3. Mechanical Properties

Table 4 shows the mechanical properties of the fiber samples. As the amount of the IFR increases, the breaking strength of the fibers decreases. According to the analysis in Section 3.2.1 regarding the aggregation structure, the addition of flame retardant will increase the crystallinity of the fibers and not significantly damage the orientation of the crystalline area for the fibers. The grooves and defects on the surface of the fibers observed from the SEM images are the main reasons for the degradation of mechanical properties. These defects are easy to generate stress concentration points under external force and cause the fibers to break. Therefore, it is not advisable to add a lot of flame retardant. The breaking strength of pure polypropylene fiber is 5.03 cN dtex^−1^. When the content of the IFR is 25%, the mechanical properties of the fiber remain at 3.35 cN dtex^−1^, showing 33.4% of reduction. However, it can still meet the strength requirements of polypropylene fabric.

### 3.3. Flame Retardant Performance and the Mode of Action for the Flame Retardant

#### 3.3.1. Flame Retardancy

Through the LOI and MCC analyses, the fire resistance performance of pure polypropylene fiber and flame retardant ones with different IFR fractions were evaluated. Figure 7 shows the LOI values and corresponding digital photos of the fibers after burning. The LOI value of pure polypropylene fiber is only 18%, and droplets are produced during the combustion process. Furthermore, there is almost no residual carbon. As the amount of IFR increases, the LOI value shows an upward trend. When the content of flame retardant is 25%, the LOI value of the fiber reaches up to 28%. The fiber has the capability of being self-extinguished when it is burned in the air. There are almost no droplets during the combustion process. Meanwhile, a significant expanded carbon layer is generated. In conclusion, the fiber with 25% Dohor-6000A have good flame retardant effect.

To further investigate the combustion behavior of flame-retardant polypropylene fibers, MCC was performed. Figure 8a,b demonstrate the curves of heat release rate with time and temperature, respectively. Table 5 summarizes the corresponding data, including pHRR, THR, heat release capacity (HRC), and the temperature corresponding to the pHRR (T_pHRR_). It is seen in Figure 8a that the pure polypropylene fiber reach pHRR in 316 s, while the flame retardant delays the time to reach pHRR. When the content of IFR is 25%, the time to reach pHRR is further extended to 366 s. The extension of time allows people to get more escape time after the fire and reduce the fire hazard. It is seen in Figure 8b that the values of pHRR and HRC of modified Dohor-6000A are very low, indicating that the flame retardant does not produce a great quantity of combustible gas during the thermal decomposition process. Compared with pure polypropylene fiber, the HRR, THR, and HRC of flame retardant fibers have decreased to varying degrees. The higher the IFR content, the greater the decrease of the HRR, THR, and HRC, indicating the less fire hazard. A high HRR peak appears in the curve of pure polypropylene fiber with a PHRR of 1042 W g^−1^ and its THR is 39.4 kJ g^−1^. When the IFR content is 25%, pHRR and THR of the flame retardant fiber drop by 26.0% and 16.0%, respectively. It shows that the flame retardant reduces the flammable gas generated during the thermal cracking of the fibers and makes the heat release rate slow. What’s more, with the addition of the IFR, the T_pHRR_ of the fibers is increased, indicating that the thermal stability of the fibers has been improved. All these results demonstrate that the modified Dohor-6000A imparts great flame retardancy to polypropylene fibers.

#### 3.3.2. Analysis of Char Residues

The pure polypropylene fiber, mPP-IFR-10 fiber, and mPP-IFR-25 fiber were burned under the condition of an open flame source in the air. Pure polypropylene fiber has almost no residual carbon, while flame retardant fibers have much residual carbon. Figure 9 is the morphology of the residual carbon for mPP-IFR-10 fiber and mPP-IFR-25 fiber. When the IFR content is 10%, the carbon layer is dense and continuous. The surface is wrinkled and there are a large number of protuberant stomates, caused by the pyrolytic gases generated by the fiber matrix and the flame retardant during the combustion process. The carbon layer covers the surface of the fiber matrix as a good barrier, which can reduce the heat transfer rate and prevent the overflow of the flammable gas generated by the fiber decomposition. It makes the fiber have good flame retardancy. When the flame retardant content is raised to 25%, it has more residual carbon, of which some is broken. The reason is that the pyrolytic gases are generated during the degradation of flame retardant [48]. More pyrolytic gases are generated with the increment of flame retardant agent fraction. When the amount of pyrolytic gases exceeds the storage capacity of char layer, the pyrolytic gases are instantly released to blow out the flame based upon the quenching effect of phosphorus-based radicals and the diluting effect of nonflammable gases. Thereby forming a broken expanded carbon layer. A similar phenomenon has also been explained [48].

From the above analysis, Dohor-6000A has an important flame retardant effect in the condensed phase for polypropylene fibers. It promotes the matrix to generate carbon layer, which has many functions such as cutting off oxygen and combustible gas, slowing down the heat transfer and decomposition of the matrix and so on.

#### 3.3.3. Pyrolysis Analysis

The thermal decomposition process of the polymer in the air atmosphere can be used as an important reference for evaluating combustion performance [49]. The TG and DTG curves of the fibers are shown in Figure 10. The relevant experimental data are listed in Appendix A When the flame retardant content is 25%, the initial decomposition temperature (T_0_) and the maximum decomposition temperature (T_max_) of the fiber are increased by 32.3 °C and 26.3 °C compared with the pure polypropylene fiber, respectively. It is noteworthy that the residual carbon rate of pure polypropylene fiber is only 0.07%, while the residual carbon rate of mPP-IFR-25 is increased to 9.22%. It shows that the IFR improves the thermal oxidative degradation stability of polypropylene fibers and increase the amount of residual carbon. From the DTG curves, the maximum decomposition rate of pure polypropylene fiber is 11.53% min^−1^, while that of mPP-IFR-25 fiber drops to 7.01% min^−1^. It shows that the IFR effectively slows down the decomposition of the polypropylene matrix.

Figure 11 is the TG-FTIR diagram of the volatile products generated by the samples during thermal decomposition. The attribution of each absorption peak is shown in Appendix A For pure polypropylene fiber, the gas generated at the beginning of decomposition (temperature below 350 °C) is mainly H_2_O (3800–3500 cm^−1^). When the temperature rises to 350 °C, the characteristic absorption peaks of various polyolefins are detected such as 2960 cm^−1^, 2840 cm^−1^, 1458 cm^−1^, 1360 cm^−1^, and 890 cm^−1^. It indicates that the main chain of polypropylene begins to decompose at this temperature. The produced combustible gases, such as alkanes and olefins, make polypropylene fiber flammable. In contrast, Figure 11b shows the FITR diagram of volatile products generated by thermal decomposition of the modified Dohor-6000A. In addition to the generation of H_2_O, NH_3_ (1620 cm^−1^, 965 cm^−1^, 930 cm^−1^) and CO_2_ (2360 cm^−1^, 2320 cm^−1^) also appear. These gases are non-combustible gases, which can dilute the oxygen during the combustion process. It can be obtained from Figure 11c,d that when the flame retardant is added, incombustible gases including H_2_O (3800–3500 cm^−1^), CO_2_ (2360 cm^−1^, 2320 cm^−1^), and NH_3_ (965 cm^−1^, 930 cm^−1^) are generated in the temperature range of 250–350 °C. The absorption peaks of alkanes and olefins for the flame retardant polypropylene fibers appear at higher temperature (380 °C) compared with that of pure polypropylene fiber (350 °C). The preferential appearance of non-combustible gases is beneficial to improve the flame retardancy of fibers.

#### 3.3.4. The Mode of Action for the Modified Dohor-6000A

The main elements of modified Dohor-6000A are carbon, nitrogen, phosphorus, hydrogen, and oxygen. The mass content of the phosphorus, nitrogen and carbon are 19.78%, 21.31% and 17.45%, respectively. Based on the above discussion, there are three main flame retardant modes of modified Dohor-6000A in polypropylene fibers: (1) In the condensed phase reaction zone, the IFR changes the pyrolysis process of the polymer molecular chain and promotes the carbonization reaction during combustion, increasing the carbon residue as a layer of glassy and stable foamy protective cover on the polymer surface. This can alleviate the internal combustion of substances. (2) The IFR will undergo endothermic reactions, thereby reducing the temperature of the surface for the polymer and the combustion zone. This makes the combustion process to be suppressed. (3) The flame retardant releases non-combustible gases such as NH_3_ and CO_2_, diluting the burning gases. The oxygen supply in the center of the flame is insufficient. Thus, a flame retardant effect is promoted.

## 4. Conclusions

In this paper, the halogen-free flame-retardant agent Dohor-6000A was successfully surface-modified and incorporated into the polypropylene matrix to fabricate melt-spun fibers with flame retardancy. The modification method improved the dispersion of the flame retardant in the blends and optimized its compatibility with polypropylene. The addition of the IFR improved the crystallization rate and crystallinity of polypropylene fibers. Adding 25% flame retardant increases the LOI value of polypropylene fiber to 29.1%. In addition, the modified polypropylene fibers hardly produced droplets during combustion. Compared with pure polypropylene fiber, pHRR, THR and HRC were decreased by 26.0%, 16.0% and 22.5%, respectively. It shows that the flame retardant reduces the flammable gas generated during the thermal cracking of the fibers and makes the heat release rate slow. The modified Dohor-6000A showed high flame-retardant activity both in the condensed phase and the gas phase. It played flame-retardant role not only by releasing non-flammable gases (H_2_O, NH_3_, CO_2_), but also by promoting the formation of a dense and expanded carbon layer.

## Figures and Tables

**Figure 1 polymers-13-02553-f001:**
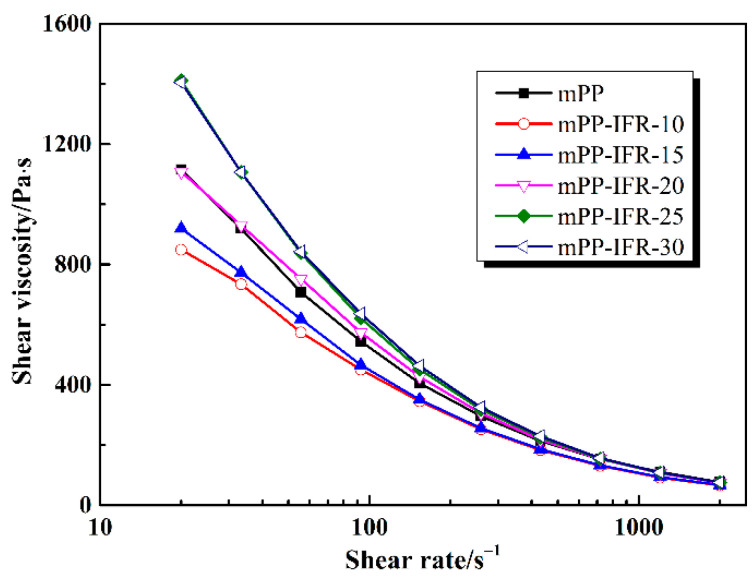
The rheology curves of polypropylene melts with different flame retardant content at 230 °C.

**Figure 2 polymers-13-02553-f002:**
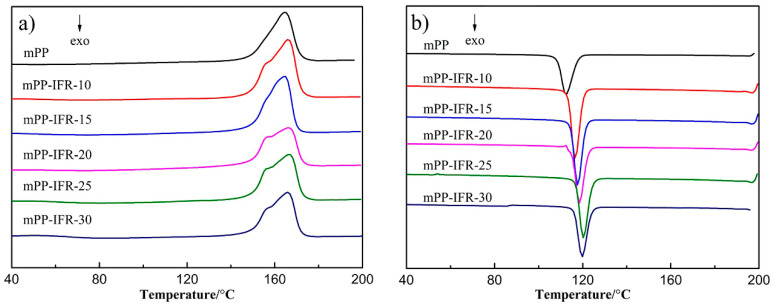
The heating (**a**) and cooling (**b**) DSC curves of the fibers.

**Figure 3 polymers-13-02553-f003:**
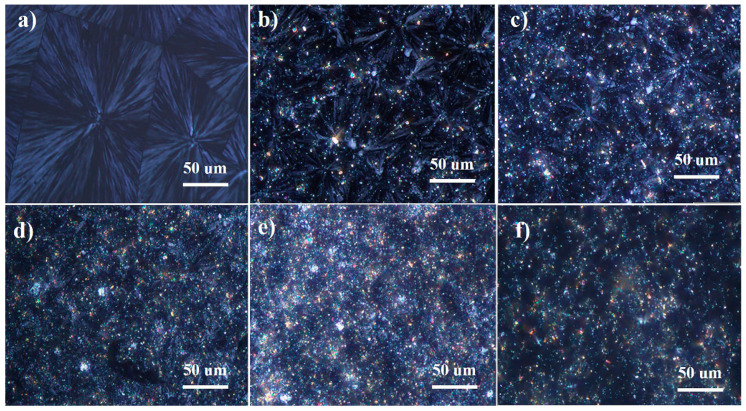
The POM photos of mPP fiber (**a**), mPP-IFR-10 fiber (**b**), mPP-IFR-15 fiber (**c**), mPP-IFR-20 fiber (**d**), mPP-IFR-25 fiber (**e**) and mPP-IFR-30 fiber (**f**).

**Figure 4 polymers-13-02553-f004:**
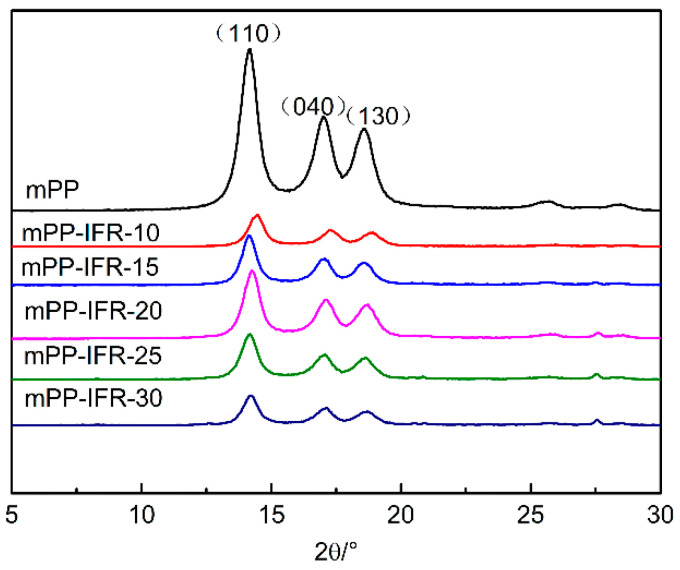
The XRD spectrum of the fibers.

**Figure 5 polymers-13-02553-f005:**
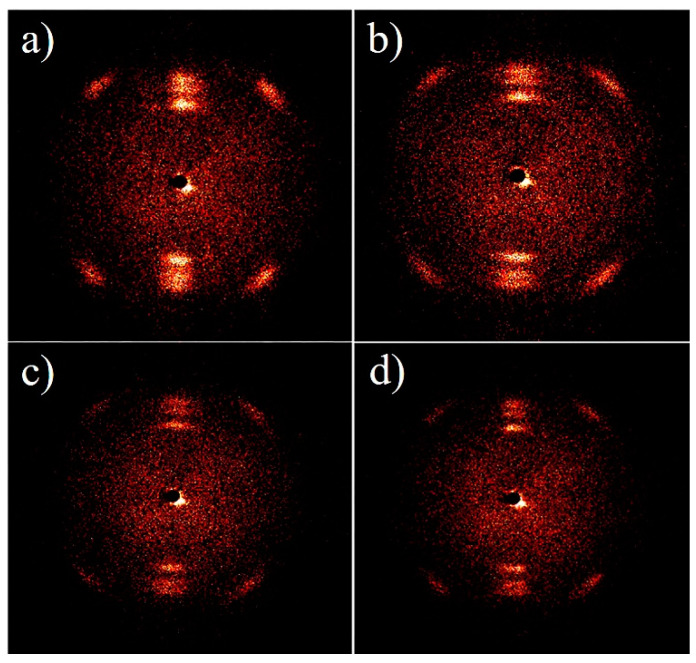
The 2D-WAXD of mPP fiber (**a**), mPP-IFR-10 fiber (**b**), mPP-IFR-20 fiber (**c**), and mPP-IFR-30 fiber (**d**).

**Figure 6 polymers-13-02553-f006:**
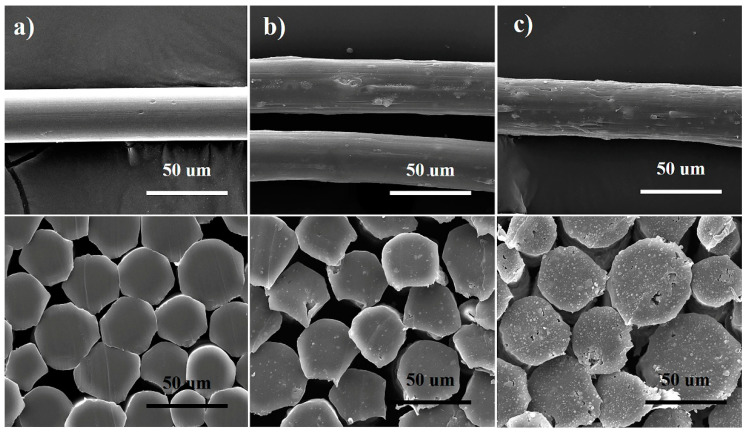
SEM images of the surface topography and cross section for mPP fiber (**a**), mPP-IFR-10 fiber (**b**) and mPP-IFR-20 fiber (**c**).

**Figure 7 polymers-13-02553-f007:**
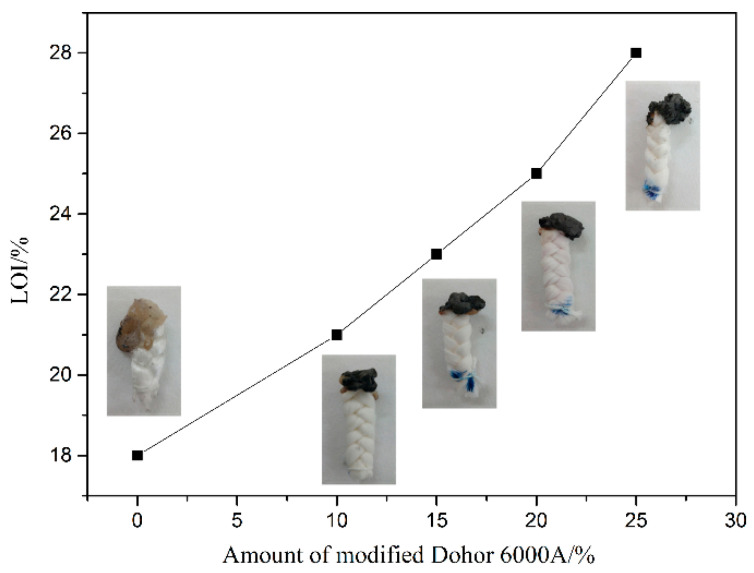
LOI values and the digital photos after LOI test of the fibers.

**Figure 8 polymers-13-02553-f008:**
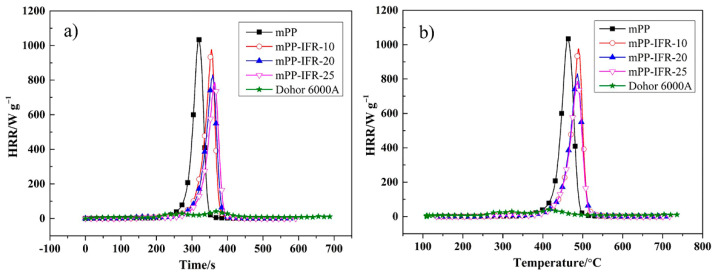
HRR-Time curve of the fibers (**a**), HRR-Temperature curve of the fibers (**b**).

**Figure 9 polymers-13-02553-f009:**
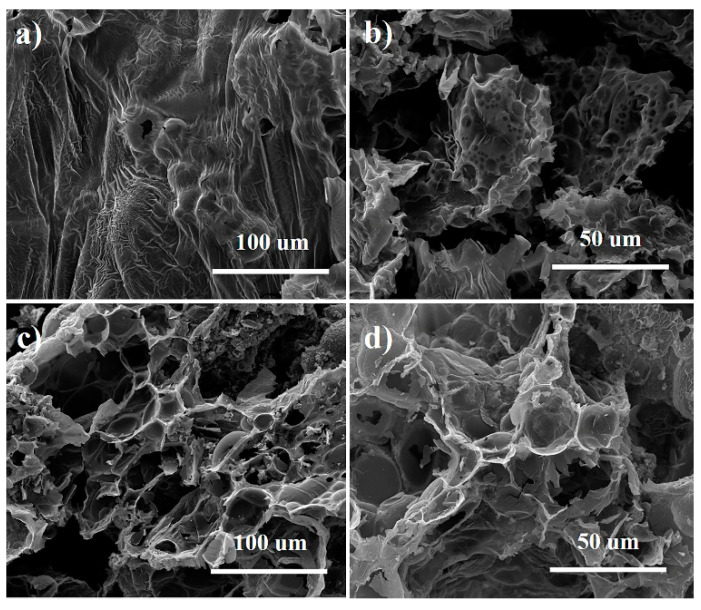
SEM photographs of char residues of mPP-IFR-10 fiber (**a**,**b**) and mPP-IFR-25 fiber (**c**,**d**) after burning.

**Figure 10 polymers-13-02553-f010:**
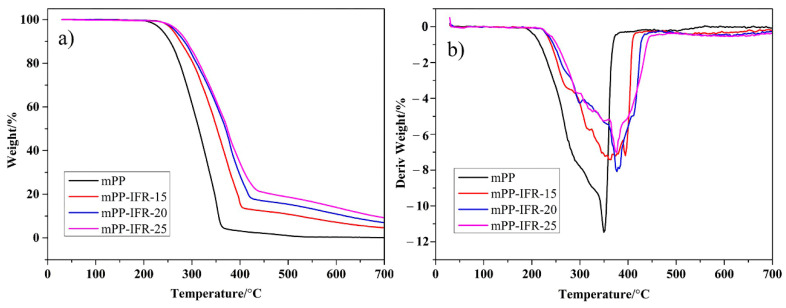
TG curves (**a**) and DTG curves (**b**) of the fibers in air atmosphere.

**Figure 11 polymers-13-02553-f011:**
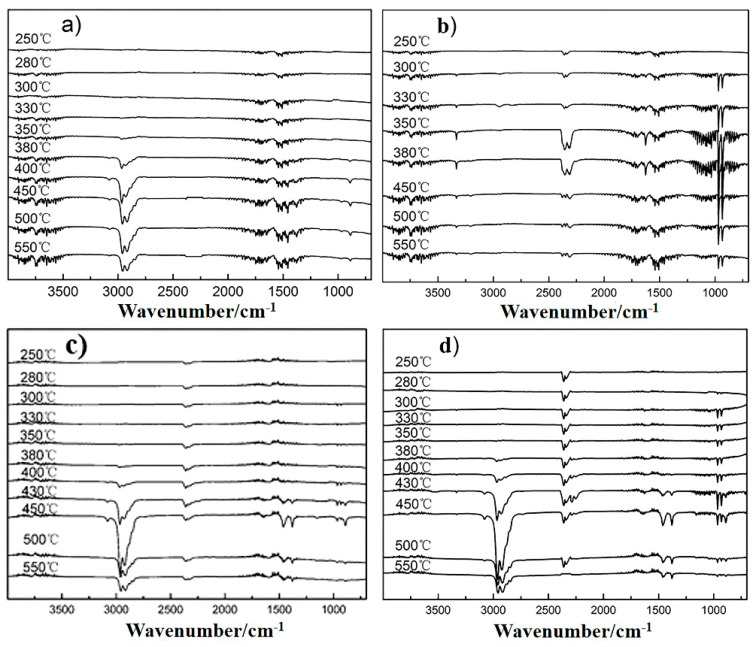
Volatile gases spectra measured by TG-FTIR for mPP fiber (**a**), modified Dohor-6000A (**b**), mPP-IFR-10 fiber (**c**) and mPP-IFR-25 fiber (**d**).

**Table 1 polymers-13-02553-t001:** The formulation and name of the samples.

Sample Code	Mass Content of Polypropylene/%	Mass Content of Surface Modified Dohor-6000A/%	Mass Content of MA-g-Polypropylene/%
mPP	95	0	5
mPP-IFR-10	85	10	5
mPP-IFR-15	80	15	5
mPP-IFR-20	75	20	5
mPP-IFR-25	70	25	5
mPP-IFR-30	65	30	5
PP-IFR-30	70	30	0

**Table 2 polymers-13-02553-t002:** The DSC data of the different fibers.

Sample	Tm/°C	Tc/°C	ΔHc/-J g^−1^	W_1/2_/°C	Xc/%
mPP	164.8	112.4	64.23	5.3	33.4
mPP-IFR-10	166.1	116.5	85.38	4.0	47.36
mPP-IFR-15	164.6	117.6	79.22	3.9	45.10
mPP-IFR-20	166.3	118.4	73.27	4.2	42.46
mPP-IFR-25	166.7	120.3	71.04	4.2	44.6
mPP-IFR-30	165.3	119.7	66.61	4.5	43.40

**Table 3 polymers-13-02553-t003:** The 2D-WAXD data of the fibers.

Sample	The Peak Width at Half Maximum (H)	Average Orientation Degree of the Crystal Region (f)
mPP	11.16	0.938
mPP-IFR-10	10.58	0.941
mPP-IFR-20	10.65	0.940
mPP-IFR-30	11.95	0.934

**Table 4 polymers-13-02553-t004:** Mechanical properties of the polypropylene fibers with different flame retardant content.

Sample	Draw Ratio	Linear Density/Dtex	Breaking Strength/cN dtex^−1^	Elongation/%
mPP	3.4	8.45	5.03	60
mPP-IFR-10	3.4	8.5	4.31	28
mPP-IFR-15	3.4	8.71	3.83	32.95
mPP-IFR-20	3.4	9.10	3.59	25.35
mPP-IFR-25	3.4	8.76	3.35	25.45
mPP-IFR-30	3.4	9.02	2.9	26.6

**Table 5 polymers-13-02553-t005:** Results of MCC tests.

Sample	pHRR/W g^−1^	HRC/J g^−1^ k^−1^	THR/kJ g^−1^	T_pHRR_/°C
mPP	1042	1034	39.4	462
mPP-IFR-10	976	980	37.7	490
mPP-IFR-20	831	840	35.0	487
mPP-IFR-25	771	801	33.1	487
modified Dohor-6000A	34.06	40	5.0	--

## Data Availability

The data presented in this study are available on request from the corresponding author.

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
