# Peer review of "Halogen-Free Flame Retardant Polypropylene Fibers with Modified Intumescent Flame Retardant: Preparation, Characterization, Properties and Mode of Action"

_polymers, 2021, doi:10.3390/polym13152553_

Round 1

Reviewer 1 Report

This manuscript describes the preparation and assessment of an intumescent flame retardant for polypropylene fiber. The introduction needs some expansion to provide a better explanation in several cases. The problems with organohalogen flame retardants is not just the formation of dioxins in a fire. A bigger problem is environmental persistence and human exposure [see Polymers, 2019, 11, 2033 - and many others - for a description]. Intumescent flame retardants are generally active in the solid phase to produce an expanded char layer at the surface of the substrate.This acts as an insulation barrier to inhibit heat feedback from the combustion zone (combustion occurs in the gas phase) which decreases the rate of polymer degradation to form volatile fuel fragments to feed the combustion process. The presence of oxygen at the surface is only important if thermooxidative processes are involved in polymer decomposition. Inert gases formed from decomposition of the blowing agent may dilute the fuel load in the gas phase but this is usually much less impactful than char formation. The phosphorus component of Dohor-6000 A is not identified (in fact, structures for none of the components are provided). If it is a phosphate (high level of oxygenation at phosphorus), it is unlikely that it provides any gas-phase activity. The nature of the interaction of the modifier with the Dohor surface is unclear. Is it simply a physical adsorption or is chemistry involved - what kind?

The manuscript will need very significant revision to make it suitable for publication. Corrections are penciled-in directly on pages of the manuscript attached. These are illustrative of the kinds of changes needed throughout. In rewriting, careful attention should be paid to the use of articles, tenses and proper sentence structure. Modification was not done "in this paper" but rather in the laboratory. Results are presented in the paper. "Mechanism" should be "mode of action". Mechanism implies far greater molecular detail than is available here. "Synergistical effects" should be removed. No evidence for synergy is provided. What is being described is a possible combination of effects (dilution in the gas phase and char formation) not synergy. Author's names and et. al. should be omitted.

Author Response

Dear reviewer,

Thank you very much for your valuable comments. In the new version of our manuscript, we have taken into account all your suggestions. And new modifications in our new manuscript were marked in yellow.

Please find below our answers:

Comment 1 from Review 1: The introduction needs some expansion.

Response: According to the reviewer's suggestions and corrections, the Title, Abstract and Introduction have been revised in detail. Some contents have been expanded including the action mode of organohalogen flame retardants and intumescent flame retardants, and the hazards of organohalogen flame retardants. Some references have been added. Relaed details were marked in yellow in our manuscript. I am very grateful for this question. Thank you very much.

Comment 2 from Review 1: The composition of Dohor-6000A and the provider of gas-phase activity.

Response: I am very sorry for not providing the structure of Dohor-6000A. In fact, the manufacturer did not provide the structure of the flame retardant, but only informed that Dohor-6000A is a phosphorus-nitrogen compound halogen-free flame retardant. After testing, we concluded that the mass-contents of the phosphorus, nitrogen and carbon are 19.78%, 21.31% and 17.45%, respectively. We are not sure whether the phosphorus component of Dohor-6000 A can provide gas-phase activity. We think the gas-phase activity may be provided by nitrogen-containing compounds. The NH3 produced during the combustion process can dilute the concentration of burning gases. I am very grateful for this question. Thank you very much.

Comment 3 from Reviewer 1: Is the nature of the interaction of the modifier with Dohor-6000A surface physical adsorption or chemistry involved?

Response: The interaction between the modifier and Dohor-6000A does not involve a chemical reaction. The modifier, KH550, undergoes a cross-linking reaction by itself under the action of water, thereby embedding the flame retardant to prevent its aggregation. In addition, maleic anhydride grafted polypropylene (MA-g-PP) is added to further promote the dispersion of the flame retardant in the matrix. The anhydride groups of MA-g-PP can attract the polar groups of KH550 on the surface of Dohor-6000A, which may form hydrogen bonds, thereby enhancing the interfacial adhesion between PP and the flame retardant. I am very grateful for this question. Thank you very much.

Comment 4 from Review 1: “Mechanism” and “Synergistical effects”.

Response: We deeply agree with the reviewer's views. I have changed “mechanism” into “mode of action”, and have removed “Synergistical effects”. I am very grateful for this question. Thank you very much.

Sincerely,

Pengqing LIU

Reviewer 2 Report

In this work the authors studied  a kind of novel intumescent flame retardant (IFR) agent named Dohor-6000A. This agent was incorporated to prepare halogen-free flame retardant polypropylene fibers. Before being blended with polypropylene resin, Dohor-6000A was surface modified firstly to improve its compatibility with polypropylene matrix. The rheological behavior of flame retardant polypropylene resin ,the aggregation structure, micro morphology, mechanical properties, fire-proof ability and related flame-retardant mechanism of the polypropylene fibers were studied in detail. X-ray diffraction (XRD) tests were also made.

Points for improvement:

  1. The results of this work could be  correlated to standards such as FMVSS 302 (Federal Motor Vehicle Safety Standards) and DIN 4102-l (Deutsches Institut für Normung) used more specifically for automotive and building sector.
  2. Please, provide every available information from the manufacturer of . Dohor-6000A. In particular, the data for the composition of Dohor-6000A is very incomplete. What kind of compounds/analysis etc.
  3. Dohor-6000A is added in high values as 30%. Is this compatible with international standards or the instructions of the manufacturer?
  4. A Google Scholar literature review made by the reviewer by using the keywords provided by the authors revealed about 30,000 works. Please, kindly complete the keywords or/and literature cited

In my opinion this work could be published after revision.

Author Response

Dear reviewer,

Thank you very much for your valuable comments. In the new version of our manuscript, we have taken into account all your suggestions. And new modifications in our new manuscript were marked in yellow.

Please find below our answer:

Comment 1: The results of this work could be correlated to standards such as FMVSS 302 (Federal Motor Vehicle Safety Standards) and DIN 4102-l (Deutsches Institut für Normung) used more specifically for automotive and building sector.

Response: I am very grateful to the reviewer for providing these two standards. FMVSS 302 can be used to describe the characteristics of horizontal burning of the materials. According to FMVSS 302, the sample size is 102 mm×356 mm×thickness (actual applied thickness, ≤13 mm). DIN 4102-l has strict requirements on the characteristics of the smoke produced by the materials during combustion. According to DIN 4102-l, different levels of testing have different requirements on sample size. This study refers to ISO-4589-2 and the sample size is 6 mm×3 mm×70 mm. These three standards have some differences in the requirements of flame retardant performance for the samples. We will try to make the flame-retardant polypropylene fibers into products for automotive and building sector. We will focus on these two standards and carry out research on products made of the flame-retardant polypropylene fibers. I am very grateful for this suggestion. Thank you very much.

Comment 2: Please, provide every available information from the manufacturer of Dohor-6000A. In particular, the data for the composition of Dohor-6000A is very incomplete. What kind of compounds/analysis etc.

Response: I am very sorry for not providing the composition of Dohor-6000A. In fact, the manufacturer did not provide the composition of the flame retardant, but only informed that Dohor-6000A is a phosphorus-nitrogen compound halogen-free flame retardant. After testing, we concluded that the mass-contents of the phosphorus, nitrogen and carbon are 19.78%, 21.31% and 17.45%, respectively. I am very grateful for this question. Thank you very much.

Comment 3: Dohor-6000A is added in high values as 30%. Is this compatible with international standards or the instructions of the manufacturer?

Response: The manufacturer recommends that the optimal amount of Dohor-6000A is between 20 wt% and 30 wt%. The maximum amount of the flame retardant added in this study reached 30 wt%, which is compatible with the instructions of the manufacturer. Considering the mechanical properties that fabric needs to meet, the most appropriate amount of the flame retardant is 25 wt%, which conforms to the instructions of the manufacturer. I am very grateful for this question. Thank you very much.

Comment 4: A Google Scholar literature review made by the reviewer by using the keywords provided by the authors revealed about 30,000 works. Please, kindly complete the keywords or/and literature cited.

Response: We carefully revise the keywords of the article. The modified keywords are polypropylene fibers, melt spinning, phosphorus-nitrogen compound, halogen-free, intumescent flame retardant and mode of action. I am very grateful for this question. Thank you very much.

Sincerely,

Pengqing LIU

Round 2

Reviewer 1 Report

This manuscript is much improved and questions have been answered satisfactorily. It will benefit from some minor corrections. At line 35, "fibers" should be fiber"; at line 39, fame-retard" should be "flame-retarding";  at line 46, "free radical" should be "radical"; at line 48, "furan" should be "furans"; at line 94, "microscop" should be "microscope", etc.

Author Response

Dear reviewer:

I am very grateful to you for earnest reviewing this article and raising some questions and suggestions. The response to the question is as follows.

Comment 1: Some words need to be corrected.

Response 1:  We have improved the English language and style of the article. Revisions in the manuscript are shown using yellow highlight. For example, at line 34, "fibers" has been changed to" fiber"; at line 38, "fame-retard" has been changed to "flame-retarding"; at line 43, "free radical" has been changed to "radical"; at line 45, "furan" has been changed to "furans"; at line 91, "microscop" has been changed to "microscope". I am very grateful for this question. Thank you very much.

Sincerely,

Pengqing Liu

Reviewer 2 Report

Dear Respected Authors,

I am happy with the revision regarding the technical part and your comments. I have noticed that the style of writting according to the other reviewer was improved only in the introduction. Although I am not a native English speaker I have the strong feeling that the manuscript has to be checked by an editing English language service. My sincere apologies for this.

Author Response

Dear Reviewer:

I am very grateful to you for earnest reviewing this article and raising some questions and suggestions. The response to the question is as follows.

Comment:

Point 1. The English language of the manuscript should be checked and corrected.

Response 1: We apologize for the language problems in the manuscript. The English language of the manuscript has been carefully checked and improved. Some words and grammar have been corrected. The sentence structure has become more reasonable. Revisions in the manuscript are shown using yellow highlight. I am very grateful for this question. Thank you very much.

Sincerely,

Pengqing Liu